# Combining the Benefits of Biotin–Streptavidin Aptamer Immobilization with the Versatility of Ni-NTA Regeneration Strategies for SPR

**DOI:** 10.3390/s24092805

**Published:** 2024-04-27

**Authors:** Eliza K. Hanson, Rebecca J. Whelan

**Affiliations:** Department of Chemistry, University of Kansas, Lawrence, KS 66045, USA; e.hanson@ku.edu

**Keywords:** surface plasmon resonance (SPR), Ni2+-nitrilotriacetic acid (NTA), biotin–streptavidin immobilization, aptamers

## Abstract

The high affinity of the biotin–streptavidin interaction has made this non-covalent coupling an indispensable strategy for the immobilization and enrichment of biomolecular affinity reagents. However, the irreversible nature of the biotin–streptavidin bond renders surfaces functionalized using this strategy permanently modified and not amenable to regeneration strategies that could increase assay reusability and throughput. To increase the utility of biotinylated targets, we here introduce a method for reversibly immobilizing biotinylated thrombin-binding aptamers onto a Ni-nitrilotriacetic acid (Ni-NTA) sensor chip using 6xHis-tagged streptavidin as a regenerable capture ligand. This approach enabled the reproducible immobilization of aptamers and measurements of aptamer–protein interaction in a surface plasmon resonance assay. The immobilized aptamer surface was stable during five experiments over two days, despite the reversible attachment of 6xHis-streptavidin to the Ni-NTA surface. In addition, we demonstrate the reproducibility of this immobilization method and the affinity assays performed using it. Finally, we verify the specificity of the biotin tag–streptavidin interaction and assess the efficiency of a straightforward method to regenerate and reuse the surface. The method described here will allow researchers to leverage the versatility and stability of the biotin–streptavidin interaction while increasing throughput and improving assay efficiency.

## 1. Introduction

Surface plasmon resonance (SPR) is a powerful label-free technique for real-time measurements of binding affinity and kinetics [1,2,3,4]. The method leverages the phenomenon that occurs when light at a given angle couples with electrons on the layer between a conducting metal and a dielectric insulator, typically gold on glass. This charged electron wave through the metal is defined as surface plasmons, and the oscillation of plasmons on the sensor surface generates an electric field [4,5,6,7]. The formation of plasmons is sensitive to the angle of incident light, which, in turn, is affected by the refractive index at the surface of the chip. When the refractive index of the sensor changes (such as upon the immobilization of a ligand), the ability of the light to cause surface plasmons is altered, allowing the researcher to quantify the binding on the surface by monitoring the intensity or shift in the angle of the reflected light [4,8]. The instrument we used in this study utilizes nanoparticles rather than planar metal to create the sensor surface. This version of the technique is called localized surface plasmon resonance (LSPR) [9,10]. When the surface comprises nanoparticles that are the same size as or smaller than the wavelength of the light, the electrons collectively oscillate and generate localized surface plasmons [9,10]. The localized plasmons have a maximum absorbance at a given resonance wavelength, and by monitoring the shift in wavelength where this maximum absorbance is observed, the binding on the surface can be measured [10]. The hybrid immobilization technique we present here would work equally well for traditional SPR; the method used to measure binding on the surface is independent of the mechanism used to immobilize the ligand.

Although SPR does not require analyte labeling, it does necessitate the immobilization of one binding partner to the surface of the sensor chip. The immobilized binding partner is typically referred to as the ligand; the binding partner free in solution is referred to as the analyte. A variety of strategies are commonly used to achieve ligand immobilization, ranging from direct approaches like thiol or carboxyl coupling to potentially reversible methods that use affinity capture tags [8]. Streptavidin–biotin is one of the most common of these capture systems for a variety of reasons (Figure 1A). Biotinylated tags are small compared to other tags such as glutathione-S-transferase (GST) and maltose-binding protein (MBP), minimizing possible interference with the structure or binding ability of the ligand [4]. Biotin can be conjugated to a variety of ligands, and the strong affinity of streptavidin–biotin binding results in an essentially permanent attachment of the biotinylated ligand to streptavidin-functionalized surfaces [4,8,11,12]. The position of the tag on the ligand offers the ability to control the orientation of the ligand relative to the sensor surface [4]. A review of 200 studies utilizing the Nicoya OpenSPR between 2016 and 2022 included 22 that opted for streptavidin–biotin immobilization [13].

Another popular strategy for immobilization is capturing ligands tagged with a poly histidine tag (6xHis) onto a nitrilotriacetic acid (NTA)-functionalized surface primed with Ni^2+^ ions [4,14] (Figure 1B). 6xHis-Ni-NTA capture has some of the benefits of streptavidin–biotin immobilization, such as small tag size and directionality of attachment, but offers the ability to regenerate the ligand surface by disrupting the complex between 6xHis tags and NTA groups by using ethylenediaminetetraacetic acid (EDTA) to remove the Ni^2+^ ions necessary for binding [4,8].

Aptamers are small, single-stranded nucleic acids that can be used as affinity reagents against a variety of targets, including proteins and small molecules [15,16]. They are smaller than antibodies, easily synthesized, and can be conjugated with a variety of modifications. Because of their fixed sequence, aptamers can be made to the exact same composition and specifications every time, and they can be shared across research laboratories by sharing a simple text string (the aptamer sequence). No such universal transfer is possible for antibodies [17,18]. Aptamers are primarily developed using a process called systematic evolution of ligands by exponential enrichment (SELEX), in which a large pool of random-sequence nucleic acids of a given length are allowed to bind to the target, separated from non-binders in the bulk library, and amplified. By performing this process multiple times, the aptamer sequences with the highest affinity for the target are separated from the pool of all potential aptamers and amplified [16,17]. Typically, the input DNA or RNA library, along with material archived during several rounds of selection, are sequenced and analyzed to determine the sequences that have become the most abundant. Often, a set of aptamers from the final round is chosen to experimentally characterize binding affinity to the target of interest. One of the most widely used model systems for protein–aptamer affinity comprises thrombin and two ssDNA aptamers referred to as the 15mer and the 29mer [18,19,20]. While both aptamers exhibit cooperative binding to thrombin with high affinity, the 29mer has been reported to have a higher affinity [17,20], though apparent affinity has been shown to depend on the method of measurement and the ionic environment [18].

While the strength of streptavidin–biotin binding is, in many ways, beneficial when immobilizing ligands of interest, it reduces the efficiency of optimizing experimental conditions. Because biotin–streptavidin binding is essentially permanent, every immobilization condition tested requires the sacrifice of a sensor chip. Even after optimizing an experimental protocol, using biotin immobilization to attach aptamers to sensor chips vastly limits the throughput of aptamer candidates an investigator can test by limiting each sensor chip to a single aptamer ligand, which is expensive and time-consuming to assess.

In this study, we report a simple strategy for immobilizing biotinylated aptamers to SPR sensing surfaces that combines the benefits of stable and reproducible biotin immobilization with the flexibility of 6xHis-Ni-NTA immobilization. The method is demonstrated using thrombin and the thrombin-binding 29mer aptamer as a model system (Figure 1C). A similar method for immobilizing proteins to measure the affinity of small molecule analytes was reported in 2021 [21], but our independent development of this technique for the assessment of aptamers adds several benefits. Our method is used to immobilize a different type of ligand target, attaching oligonucleotides to the sensor rather than proteins. By immobilizing the smaller binding partner of interest (the aptamer), the possible binding signal from the larger protein analyte is maximized. In addition, the two-reagent strategy for ligand regeneration we report is simple and quickly removes the ligand, enabling surface preparation and further ligand immobilization with fewer steps and reagents required than the protocol they report for regenerating immobilized proteins off the surface. While Gunnarsson et al. note that ligand regeneration will depend on the ligand of interest [21], it is likely that different aptamer candidates will behave similarly when regenerating aptamer-6xHis–streptavidin complexes off the surface. They report the immobilization of proteins that contained both 6xHis and biotin tags, which does not allow for clarity whether the mechanism of immobilization occurs via the biotin tag binding to streptavidin protein on the sensor or due to the 6xHis tag on the protein. Our method verifies that immobilization results from the specific interaction of the biotin tag and the streptavidin protein by demonstrating binding with a complementary binding affinity technique and by using ligands that contain only a singular capture tag. This demonstrates that ligands do not require a 6xHis tag in order to use this hybrid immobilization method.

In addition to reproducible immobilization and binding affinity measurements, the stability of the aptamer-immobilized surface over time was tested. Binding affinity characterizations between an immobilized aptamer and five sets of thrombin samples over the course of two days were reproducible, giving researchers the opportunity to confidently use the same surface for multiple experiments to measure several protein targets or collect sets of replicate measurements. While the present study only tested the stability of an immobilized aptamer surface through two full days of experiments, it is probable that the surface could have been used for additional experiments. Finding the limit of exactly how long the same aptamer-immobilized surface can be used for binding affinity assays will require further experimentation. In terms of immobilization reproducibility rather than immobilization stability, Nicoya has demonstrated the ability to reproducibly immobilize 6xHis–streptavidin up to 27 cycles and measure 6xHis protein A–IgG binding for 18 cycles in their NTA sensor ligand recapture on OpenSPR^TM^ tech note [22]. The ability to perform many different reproducible immobilizations, along with the ability to use individual aptamer-immobilized surfaces for multiple assays, provides investigators with a great deal of flexibility while designing experiments.

## 2. Materials and Methods

### 2.1. Materials

Experiments were performed on an OpenSPR-XT 2-channel instrument from Nicoya Lifesciences (Kitchener, ON, Canada) with NTA-functionalized sensors and well plate foils from the same vendor. In the two-channel set-up, the first channel (Ch1) serves as a reference and the second (Ch2) is where ligand is immobilized. The calculated corrected signal corresponds to the response from Ch1 subtracted from Ch2 to measure only the interaction of the bound ligand and the free analyte, rather than any non-specific binding on the surface.

Lyophilized bovine serum albumin (BSA) was purchased from MilliporeSigma (Burlington, MS, USA) and reconstituted in water to a concentration of 1 mg/mL. Lyophilized thrombin derived from human plasma was purchased from MilliporeSigma and reconstituted to 1000 U/mL in 1 mg/mL BSA. 6xHis tagged streptavidin (His-SA) at a stock concentration of 1 mg/mL in 20 mM Tris-HCl pH 7.5 was purchased from Fitzgerald Industries, (Acton, MA, USA), My Biosource (San Diego, CA, USA), and Abcam (Waltham, MA, USA). Aptamers were purchased from Integrated DNA Technologies (Coralville, IA, USA) both with a biotin tag (B-29mer) and without (unlabeled 29mer) on the 5′ end (sequence: /5Biosg/AG TCC GTG GTA GGG CAG GTT GGG GTG ACT). Lyophilized aptamers were reconstituted in 1X Tris EDTA (TE) buffer (VWR, Radnor, PA, USA) to a concentration of 100 µM and stored at −20 °C. NiCl_2_ was purchased from BeanTown Chemical (Hudson, NH, USA) and reconstituted to 40 mM in water. EDTA was purchased from TCI America (Portland, OR, USA). Glycine, D-(+)-Biotin, NaCl, NaOH, and HCl were all purchased from VWR. A solution of 10 mM glycine–HCl was prepared from solid glycine and pH adjusted to 1.5 with HCl. The 10X phosphate-buffered saline (PBS) and Tween 20 were purchased from Thermo Fisher (Waltham, MS, USA). We then prepared two modified buffers in house for our experiments. PBS-T was 1X PBS and 0.05% Tween 20 (*v*/*v*) and used as the sample buffer. Some experiments also utilized an immobilization buffer (PBS-I) that was 1X PBS with NaCl added to a total concentration of 300 mM, with pH adjusted with NaOH to a final pH of 8.0. Data were analyzed using TraceDrawer Analysis 1.9.2 (Ridgeview Instruments, Uppsala, Sweden) and IGOR Pro (WaveMetrics, Portland, OR, USA).

### 2.2. General Experimental Overview

In brief, each experiment entailed cleaning the surface of the sensor chip with HCl and EDTA, followed by priming the surface with NiCl_2_ and immobilizing His-SA into both channels. B-29mer was then immobilized into only channel 2, and a range of concentrations of thrombin protein were injected to measure the protein–aptamer binding signal. Average corrected curve signals were calculated from a given interval towards the end of the injection and plotted versus concentration in order to fit the data with the Hill equation:f_b_ = (B_max_*[P]^n^)/(K_A_^n^ + [P]^n^).(1)
Then, we calculated the binding constants, as described by Mears et al. [18], where the fraction bound is represented as f_b_, B_max_ is a fit parameter for the maximum observed signal, the n coefficient accounts for cooperative binding, protein concentration is given as P, and K_A_ is a dissociation constant that corresponds to the concentration of protein at half the maximum binding signal. Specific experimental details that sometimes differed between experiments can be found in Table 1.

### 2.3. Demonstration of a Typical Binding Affinity Assay (Experiment 1)

B-29mer, His-SA, and thrombin samples were diluted in PBS-T sample buffer. B-29mer was diluted into PBS (to avoid heat degradation of Tween20) at a concentration of 10 µM and heated to 95 °C for 3 min then cooled to 4 °C and held at that temperature until further sample preparation. B-29mer was then diluted to a final concentration of 1 µM in PBS-T. His-SA was prepared to a concentration of 1 µM. Thrombin samples were prepared from stocks reconstituted in 1 mg/mL BSA, and an appropriate volume of 1 mg/mL BSA was added to each sample so that the volume contributed between thrombin and stock BSA in each was 10 µL. Samples of thrombin were then diluted in 200 µL PBS-T to their final concentrations, which ranged from 5 to 400 nM.

Unless otherwise noted, all injections used a flow rate of 20 µL/min, a dissociation time of 270 s, and were injected into both channels. The sensor surface was cleaned with two injections of 10 mM glycine–HCl, pH 1.5 (at 150 µL/min), and one injection of 350 mM EDTA (at 100 µL/min). The NTA surface was then primed with 40 mM NiCl_2_, and His-SA was immobilized into both channels. B-29mer was then injected into channel 2 only. Both His-SA and B-29mer had extended dissociation times of 600 s each. Following immobilization, injections alternated between regeneration using 2 M NaCl (at 150 µL/min) and thrombin analyte injections (concentration order was scrambled across the test). At the end of the test, an additional NaCl regeneration step along with two glycine–HCl injections and one EDTA injection were performed to fully remove the ligand from the surface and clean the sensor for future experiments.

The data were exported into TraceDrawer and the average signal for each curve over the interval between 475–500 s was calculated. The average curve interval for each concentration was plotted in IGOR Pro and fit with the Hill equation using values of n calculated from a Hill plot of the data.

### 2.4. Immobilization Reproducibility (Experiment 2)

His-SA and B-29mer were prepared as for Experiment 1 but at triple the volume to account for 3 separate immobilizations onto the same chip. Representative thrombin concentrations of 10, 100, and 400 nM were prepared as in Experiment 1, performing the initial dilution in the appropriate volume of 1 mg/mL BSA so that the background contribution to each sample was the same (with tripled volume to allow for three replicates). Immobilization of His-SA and B-29mer used the same parameters as Experiment 1. Following immobilization, injections alternated between 2 M NaCl to regenerate the surface and thrombin samples. After the last thrombin sample, the surface was regenerated with NaCl and EDTA. This protocol was repeated in triplicate, starting with two glycine–HCl injections and one EDTA injection to ensure the surface was fully regenerated before reimmobilizing His-SA and B-29mer.

The data were exported into TraceDrawer and the average signal for each curve over the interval of 800–825 s was calculated for the corrected B-29mer signals and both His-SA channels 1 and 2.

### 2.5. Experimental Reproducibility (Experiments 3 and 4)

The same parameters as above were used to prepare and immobilize His-SA and B-29mer onto a fresh NTA chip. Thrombin concentrations ranging from 10 to 300 nM were prepared (using the same strategy as earlier experiments with regards to BSA dilutions), and each was prepared with enough volume for 3 analyte injections. In addition, a sample of 0 nM protein was prepared (with the same volume of BSA as samples) to assess the non-specific binding of samples due to the presence of BSA. The instrumental protocol used the same strategy for surface cleaning and B-29mer immobilization, but a single B-29mer-immobilized surface was used for all three sets of thrombin analytes. Following immobilization, each set of thrombin samples was injected in a scrambled order with NaCl injected between thrombin samples to remove analyte from the surface. Once a full set of thrombin concentrations had been run, an additional injection of NaCl was used to ensure analyte was fully removed from the surface, and the next set of analytes was injected. At the end of the test, His-SA and B-29mer were regenerated off the surface using two glycine–HCl injections and one EDTA injection, and the sensor chip was put into standby overnight. Experiment 4 used an almost identical protocol as above but included a 5 nM thrombin sample in each set of analytes.

The average analyte signal for each curve was calculated over the interval of 475–500 s, and the average signal of each day’s three replicates was plotted versus the concentration and fit with the Hill equation on IGOR Pro.

### 2.6. Immobilization Stability (Experiments 5–8)

Experiments 5 and 6 were an initial scouting run to assess the feasibility of using the same immobilized surface on separate days. Experiments 7 and 8 were similar protocols run with triplicate and duplicate sets of samples, respectively.

Experiments 5 and 7 followed similar sample preparation and instrumental protocols as Experiments 1–4, with a few differences. PBS-I was used to prepare the samples for immobilization (B-29mer, His-SA, and an added injection of 255 µM biotin as a blocker), and they were each injected at 20 µL/min with a dissociation time of 270 s. B-29mer and His-SA concentrations were prepared in excess, i.e., to 10 µM and 2.5 µM, respectively. Following biotin injection, instrumental protocol incorporated a buffer swap from PBS-I to PBS-T for the rest of the run. Thrombin samples were prepared in PBS-T, with the same strategy to standardize BSA volume across samples. Thrombin samples were prepared fresh each day; Experiments 5 and 6 each had one replicate of each analyte concentration and Experiments 7 and 8 had enough sample prepared for triplicate injections of each analyte concentration. Instrumental protocols used the same parameters as earlier experiments but did not include the regeneration injections to remove His-SA and B-29mer from the surface of the chip at the end of Experiments 5 and 7. Following the experiment on the first day, the instrument was put into standby overnight with the sensor temperature set to 12 °C before preparing and running the next set of analytes the following day on the same immobilized surface. Data were analyzed as in Experiments 3 and 4.

## 3. Results

### 3.1. Demonstration of a Typical Binding Affinity Assay (Experiment 1)

While the primary function of His-SA in this system is to create a surface to immobilize biotinylated ligands, it also serves as a blocking molecule in the reference channel to prevent non-specific interactions of B-29mer with the NTA chip surface. The sensor surface is cleaned at the start of a test, followed by an injection of a high concentration of His-SA, which shows strong a immobilization signal in both channel 1 and channel 2 (Figure 2). Following His-SA immobilization, B-29mer is injected into channel 2, binding to available streptavidin sites on the surface (Figure 2).

A gel shift assay between B-29mer with a biotinylated tag and unlabeled 29mer with His-SA protein demonstrated that binding was only observed between B-29mer and His-SA; thus, immobilization is due to specific interaction between B-29mer and the His-SA on the surface rather than non-specific adsorption (Appendix A). While BSA at high concentrations can demonstrate non-specific binding to the 29mer, blank samples (BSA and sample buffer, but no thrombin present) demonstrate a negligible signal during an SPR assay (Appendix A). In addition, the contribution of BSA to each sample is standardized, which should eliminate any effect on binding due to the subtraction of reference channels from corrected binding signals.

Following successful immobilization of B-29mer, the binding of a series of thrombin concentrations was measured. Sensorgrams of the corrected signal for each thrombin concentration are shown in Figure 3A, with the binding isotherm shown in 3B. Thrombin binds rapidly to the aptamer immobilized on the surface, as seen in the association phase as protein is injected into the flow cell for the first 300 s. Following analyte injection, the instrument switches to running buffer, and the slope trends slightly downwards as protein slowly dissociates off the aptamer.

By calculating the average signal for each curve over the same interval, we can plot the average signal versus concentration and fit it with the Hill equation (Figure 3B). The K_A_ calculated for this test is 31.2 ± 1.0 nM, a value comparable to the K_A_ values reported for the same system measured via fluorescence anisotropy and affinity probe capillary electrophoresis [18].

### 3.2. Immobilization Reproducibility (Experiment 2)

We observed reproducible immobilizations of both His-SA protein (Figure 4A) and B-29mer (Figure 4B) when we compared triplicate immobilization cycles performed on the same chip.

The immobilization level of His-SA was comparable for both the reference channel and the ligand channel prior to B-29mer immobilization, and the corrected signal of B-29mer was similarly reproducible across three repeated immobilizations. The coefficients of variation for each set of replicates ranged from 3.9% to 9.5%, as reported in Table 2, validating the use of this system to immobilize multiple biotinylated ligands to the same chip.

We also assessed the efficiency of our regeneration method utilizing two glycine–HCl injections and one EDTA injection. By comparing the average curve interval for 15 s windows at the end of an immobilization step and following regeneration steps, we found that regeneration for His-SA alone off the surface was >96% (measuring channel 1), and the regeneration for His-SA bound to B-29mer was >95% (assessed using channel 2) for all three immobilizations.

### 3.3. Experimental Reproducibility (Experiments 3 and 4)

Once the reliability of multiple immobilizations on the same chip had been shown, we wanted to assess the reproducibility of experiments taken as a whole. We observed comparable behavior for two separate immobilized surfaces on the same chip on separate days in Experiments 3 and 4 (Figure 5).

Incomplete analyte regeneration off the surface can cause a drop in the measured binding of subsequent samples due to surface saturation, particularly over the course of long experiments with many sample injections. To verify successful regeneration, the return of the signal to baseline was monitored between analyte injections and over the course of the full test. The full test preview of Experiment 3, showing successful analyte regeneration and a steady baseline throughout the test, is found in Appendix A.

The calculated K_A_ values for Experiment 3 and 4 were 9.1 ± 0.4 and 9.6 ± 0.6 nM, respectively. The analyte binding was consistent within experiments conducted on each day’s surface, as shown by the low variability in the standard error of the mean for each point. Analyte response was also consistent between the two different days, as demonstrated via similar calculated K_A_ values.

### 3.4. Immobilization Stability (Experiments 5–8)

The reproducibility of replicate ligand immobilizations on a chip lends confidence to individual assays as variability in ligand immobilization can cause variation in measurements of analyte binding [23]. The ability to use a chip for multiple immobilizations increases throughput. However, being able to use the same surface multiple times before regeneration would be beneficial for several reasons, including the conservation of precious samples as well as saving time by running more analyte samples before needing to repeat immobilization steps. Biotin–streptavidin immobilization enables stable binding, and we wanted to establish that this stability was not lost when biotin–streptavidin coupling was combined with reversible immobilization on an NTA chip. Figure 6 demonstrates that similar signals are obtained from thrombin samples run on the same immobilized B-29mer surface on the second day after leaving it on standby overnight in the instrument. Experiments 5 and 6 were an initial scouting experiment undertaken to determine the feasibility of using the same immobilized surface on two separate days, and very similar binding signals can be seen in Figure 6A.

The analyte binding measured on the same surface was similar for both days, with a calculated K_A_ of 17.6 ± 1.6 nM for Experiment 5 and 16.7 ± 3.8 nM for Experiment 6.

In order to verify the results of Experiments 5 and 6, the number of replicates run on a chip was increased. Over the course of two days, there was comparable binding from analyte samples through five full sets of thrombin samples across the same B-29mer surface. Isotherms comparing each day’s pooled average curve signals is shown in Figure 7.

The results from these experiments affirm what the preliminary scouting run of Experiments 5 and 6 seemed to indicate—an immobilized aptamer surface can be used for more than one day. We observed reproducible analyte binding measured across the surface through five runs over two days. The calculated K_A_ for Experiment 7 was 18.6 ± 2.8 nM, and the K_A_ for Experiment 8 was 10.3 ± 0.4 nM. The hybrid chip immobilization design shown in this paper retains the strength and stability of a biotin–streptavidin chip format while benefiting from the flexibility and versatility of an NTA system.

### 3.5. Hybrid chip Design Is Robust and Reproducible across Different Chips and Experimental Parameters

In addition to comparing the reproducibility of immobilizations and analyte binding on the same chip, we have collected the calculated binding constants for all the experiments in this study (Table 3).

While the K_A_ values for all the experiments are not identical, we know from Mears et al. that a wide range of K_A_ and K_d_ values have been reported for the thrombin binding aptamers across buffer conditions and between experimental methods [18]. The K_A_ for Experiment 1 at 31.2 nM was the highest, but the spread of binding constants calculated fall within the expected range for thrombin–aptamer binding, both within the literature values and compared to the typical results we see in house for this experiment [18]. The main difference between the K_A_s calculated in this paper and the data observed by Mears et al. [18] is that the SPR results tend to have a lower K_A_ on average, but this is in keeping with their observation that different experimental methods measure slightly different bindings. The two main methods reported by Mears et al. [18] are fluorescence anisotropy and affinity probe capillary electrophoresis, which measure analyte binding free in solution. Because our values are close to those reported in the literature (with slightly higher affinity, in most experiments), this increases the confidence in our immobilization method. The immobilization of an aptamer on the sensor chip surface for up to two days does not affect the ability to reproducibly measure the binding affinity between thrombin and the thrombin-binding aptamers, reinforcing that the surface is stable and reliable.

There is a much wider range in the calculated B_max_ values, but this is likely due to differences in sensor chip surfaces and immobilization efficiency [23], and experiment-to-experiment variability between the B_max_ values does not affect the relative consistency of the calculated K_A_s.

## 4. Conclusions and Future Directions

Aptamer selection has been aptly compared to finding a needle in a haystack. The nature of this process requires investigators to characterize the binding of a set of promising aptamer candidates, rather than a single candidate. Because of its ability to measure analyte targets without labels and the small size of a biotin tag, which minimizes potential interference, SPR is, in many ways, an ideal method of characterizing aptamer–target binding. However, the throughput is severely limited on traditional streptavidin–biotin chips, which require a new sensor for each aptamer candidate. The hybrid immobilization method reported here retains these benefits while allowing investigators to immobilize and characterize multiple aptamer candidates on a single chip. While this hybrid immobilization method does require an additional step to achieve, it removes the time required to clean and dock new sensor chips from the workflow. In terms of total experimental run time, the two immobilization methods are largely comparable. For applications that are high throughput like aptamer characterization, the extra step is a worthwhile tradeoff for the ability to run multiple aptamers on the same sensor chip. In addition to reducing the number of sensor chips required, it can reduce variability that might occur between separate chips or separate sensor lots [23]. In future work, we would like to assess the immobilization of other aptamers, particularly ones of different lengths or with affinity for different types of targets. Using a biotinylated library (or pools amplified with a biotinylated forward primer) to assess individual rounds throughout the selection would provide a rapid and informative method by which to perform bulk affinity. One of the difficulties of SELEX is determining the optimum number of rounds of selection; too few rounds can result in weakly binding aptamers or a failed selection, but too many rounds wastes time and allows easily amplified but off-target aptamers to accumulate through PCR. By assessing bulk affinity quickly after each round, this would allow for a more confident determination of when to end selection.

## Figures and Tables

**Figure 1 sensors-24-02805-f001:**
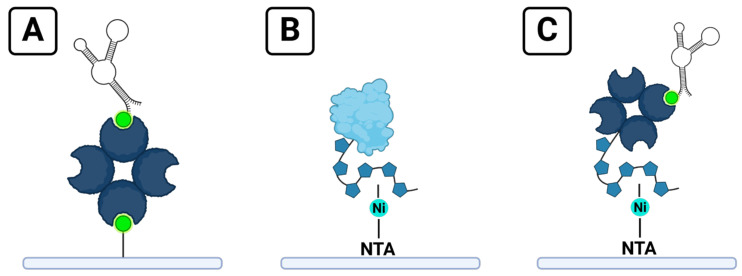
Strategies for ligand immobilization on SPR sensor chips: (**A**) a typical streptavidin–biotin chip; (**B**) immobilization of a protein ligand with a 6xHis tag onto a Ni-NTA chip; (**C**) the hybrid chip format described in this manuscript, combining a biotinylated ligand with the reversible immobilization of streptavidin onto a Ni-NTA chip.

**Figure 2 sensors-24-02805-f002:**
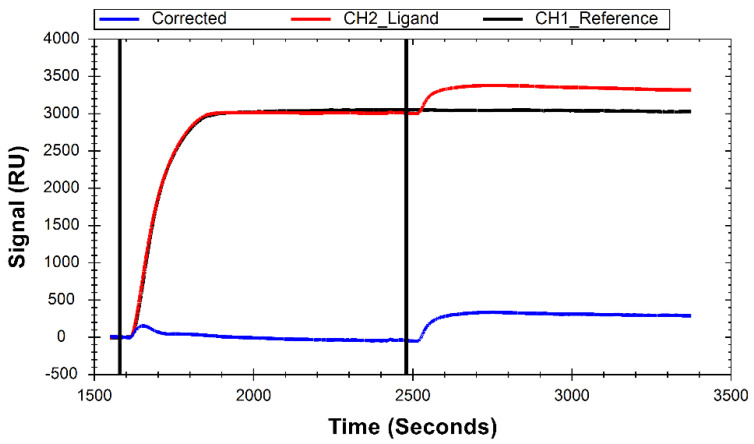
Sensorgram demonstrating immobilization of B-29mer onto H-SA. Curves were cropped from full test preview signal and aligned to Y = 0 at the start. The black line at 1595 s indicates the injection of H-SA into both channels, and the line at 2480 s indicates the injection of B-29mer into channel 2.

**Figure 3 sensors-24-02805-f003:**
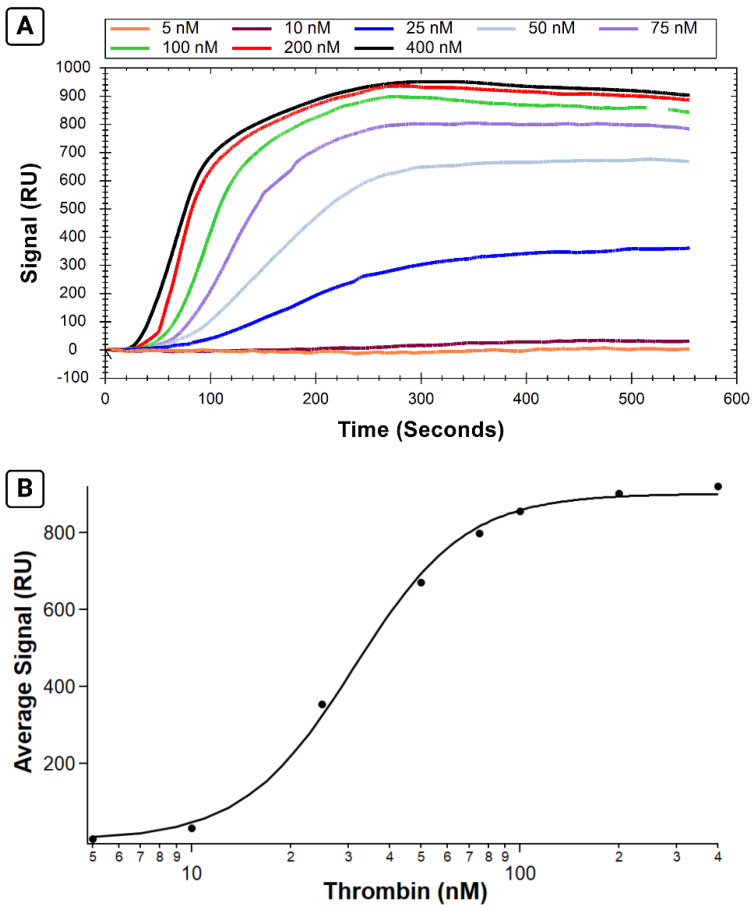
Results from SPR assay measuring binding affinity of B-29mer and thrombin. Sensorgram signals from the corrected channel shown in (**A**), with the x-axis for all curves trimmed at 555 s. The air bubble spike was trimmed from the 100 nM concentration trace from 515 to 535 s. An isotherm plotting the average signal from each concentration is shown in (**B**). The data were fit with the Hill equation.

**Figure 4 sensors-24-02805-f004:**
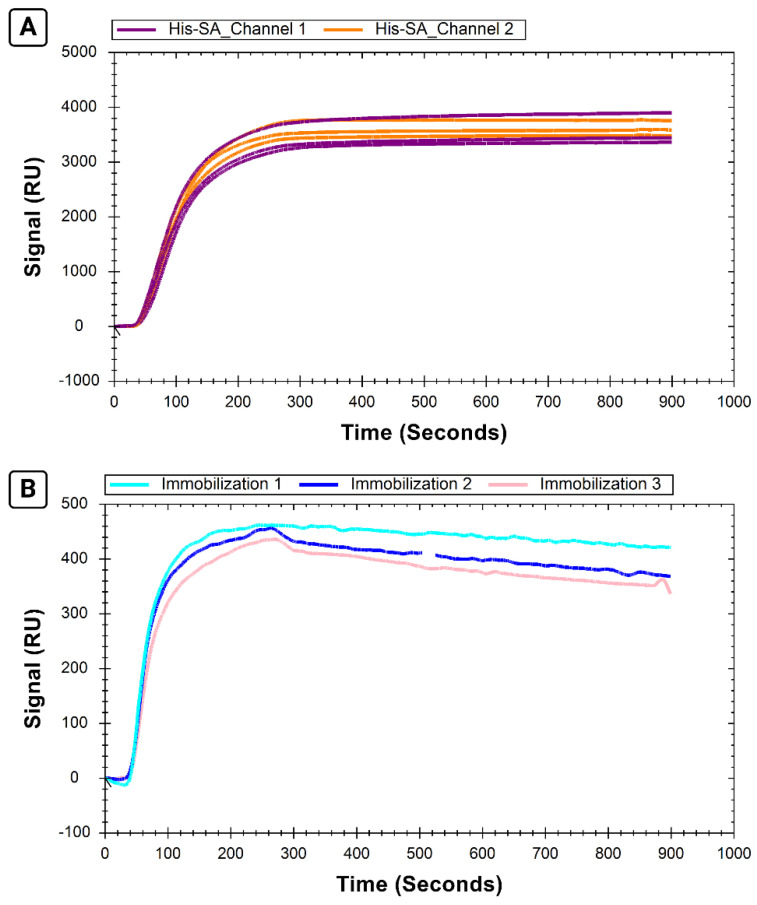
Sensorgrams showing replicate immobilizations of B-29mer and H-SA. The signals from both Ch1 and Ch2 for H-SA are shown in (**A**); the corrected signal shown for B-29mer immobilization is shown (**B**). Air bubble spike removed from the second B-29mer injection from 505 to 525 s.

**Figure 5 sensors-24-02805-f005:**
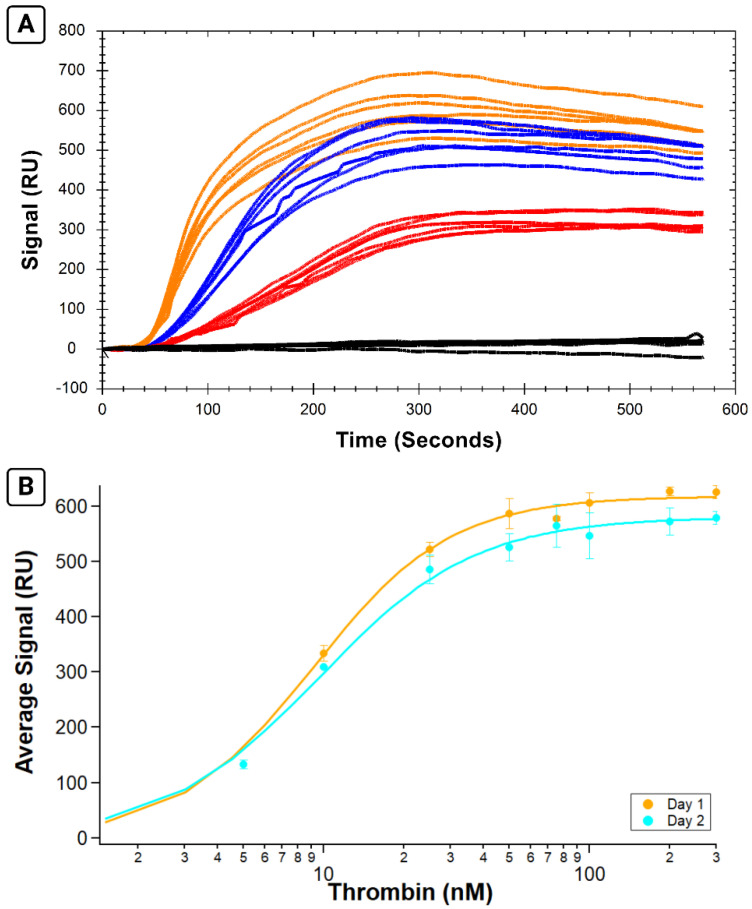
Six replicate binding affinity assays measuring B-29mer and thrombin on two different immobilized sensor surfaces on the same sensor chip. Selected sensorgram traces colored according to thrombin concentration in (**A**), with black traces corresponding to 0 nM point, red traces denoting 10 nM samples, blue traces for 25 nM, and the orange signals from the 75 nM samples. (**B**) The three replicates per day were averaged and plotted on an isotherm then fit with the Hill equation. Experiment 3 on the first day’s immobilized surface is shown in orange, and Experiment 4 on the second day’s freshly immobilized surface is shown in teal. Error bars are standard errors of the mean.

**Figure 6 sensors-24-02805-f006:**
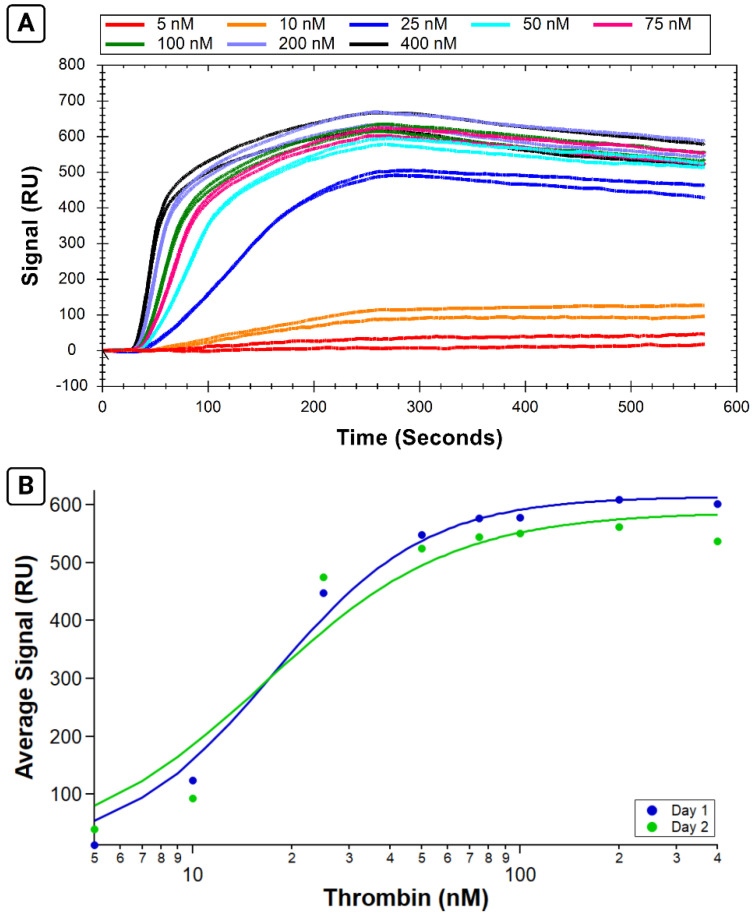
Comparison of thrombin samples run on the same B-29mer-immobilized surface on two consecutive days. Sensorgrams plotting the thrombin binding signals are given in (**A**), with curves color-coded by concentrations. Each day’s experiment was plotted on IGOR and fit with an isotherm, as shown in (**B**). Experiment 5 (run prior to putting the immobilized surface on standby overnight) is in dark blue and Experiment 6 (run the next morning) is in green.

**Figure 7 sensors-24-02805-f007:**
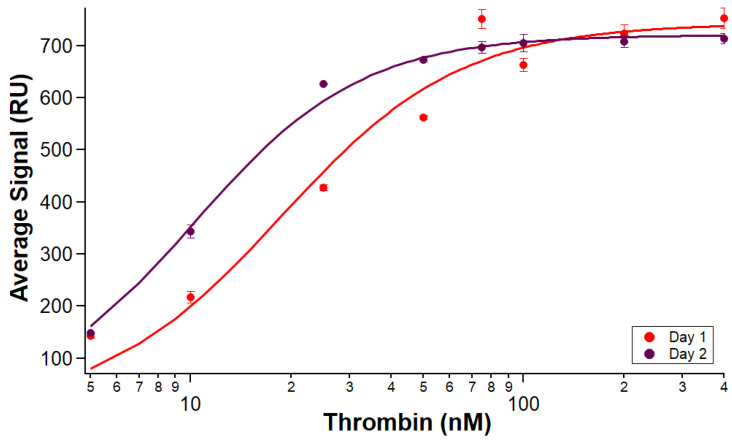
Five replicate binding affinity tests on the same immobilized surface across two days. B-29mer was immobilized at the beginning of the test. Sets of thrombin samples were injected on the surface, and binding was measured. Experiment 7 on the first day (red) had triplicate samples, and Experiment 8 on the second day (maroon) had duplicate samples. Error bars are standard errors of the mean. One measurement of thrombin at 5 nM on the first day was omitted because of air bubble spikes.

**Table 1 sensors-24-02805-t001:** Experimental parameters.

Expt. No.	Section	[B-29mer]	[His-SA]	No. Replicates	Immob. Buffer	Range of [Thrombin]
1	Typical binding affinity assay	1 µM	1 µM	1	PBS-T	5–400 nM
2	Immobilization reproducibility	1 µM	1 µM	3	PBS-T	10–400 nM
3	Experimental reproducibility	1 µM	1 µM	3	PBS-T	0–300 nM
4	Experimental reproducibility	1 µM	1 µM	3	PBS-T	0–300 nM
5	Immobilization stability	10 µM	2.5 µM	1	PBS-I	5–400 nM
6	Immobilization stability	no immobilization		1	none	5–400 nM
7	Immobilization stability	10 µM	2.5 µM	3	PBS-I	5–400 nM
8	Immobilization stability	no immobilization		2	none	5–400 nM

**Table 2 sensors-24-02805-t002:** Immobilization reproducibility.

	B-29mer	His-SA Ch1	His-SA Ch2
**Average RU**	386.44	3558.52	3606.90
**CV**	9.5%	8.1%	3.9%

**Table 3 sensors-24-02805-t003:** Calculated binding constants for each experiment given and the number of replicates.

Experiment No.	No. Replicates	K_A_ (nM)	B_Max_ (RU)
1	1	31.2 ± 1.0	902.7 ± 11.2
2	Primarily immobilizations; no K_A_ and B_Max_ to report		
3	3	9.1 ± 0.4	617.9 ± 6.6
4	3	9.6 ± 0.6	580.4 ± 0.7
5	1	17.6 ± 1.6	614.7 ± 15.6
6	1	16.7 ± 3.8	588.0 ± 34.5
7	3	18.6 ± 2.8	742.8 ± 29.6
8	2	10.3 ± 0.4	720.5 ± 7.3

## Data Availability

Data are contained within the article.

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
