# Peer review of "Combining the Benefits of Biotin–Streptavidin Aptamer Immobilization with the Versatility of Ni-NTA Regeneration Strategies for SPR"

_sensors, 2024, doi:10.3390/s24092805_

Round 1

Reviewer 1 Report

Comments and Suggestions for Authors

In this work the authors propose a hybrid ligand immobilization procedure combining a biotinylated analyte with the reversible immobilization of streptavidin onto a Ni-NTA chip. Please find below my comments:

1. Despite a good context is given, the abstract should better focus on the main findings of the proposed work;

2. The introduction part relative to LSPR (lines from 35 to 42) appears out of context and it could be removed.

3. The authors should clearly state the novelty of the proposed work with respect to the state-of-the-art (e.g., Ref. 18).

4. In the Introduction Section the authors state "...giving researchers the opportunity to confidently use the same surface for multiple experiments to measure several protein targets or collect sets of replicate measurements." Have the authors determined how many times the sensor chip is reusable without causing any kind of surface degradation?

5. The hybrid immobilization procedure investigated by the authors appears more laborious with respect to the typical Streptavidin–Biotin one. Could it be a considered a good trade-off in the view of sensor chip reusability? Please give a comment about this aspect.

6. To improve the readability of the paper, the authors should include the Hill equation they used to fit the data.

7.With regards to experimental reproducibility, the authors found similar Ka values in two different days. However, this Ka values (equal to around 9 nM) were quite different from the one found in Section 3.1 (around 31 nM). Can the authors clarify this aspect and give a possible explanation to this difference?

8. I suggest the authors to use a semi-log scale for Figs 3B, 5B, 6B and 7.

Reviewer 2 Report

Comments and Suggestions for Authors

The authors have introduced a method to reversibly immobilize biotinylated thrombin-binding aptamers onto a Ni-nitrilotriacetic acid (Ni-NTA) sensor chip using 6xHis-tagged streptavidin as a regenerable capture ligand. While the article is well-written and the discussions are engaging, several points need to be addressed before accepting this manuscript with minor revisions.

1.      In the introduction section, please briefly elaborate on the novelty of the work and clarify the unique properties of the proposed structures compared to existing literature.

2.      Please include more references related to SPR effects (DOI: 10.1016/j.rinp.2019.102290 and DOI: 10.1007/s11051-016-3394-1) to support the description on lines 25-26.

3.      Please investigate why the blue curve in Figure 4(B) exhibits discontinuous breaks.

4.      The mechanism and description of Figure 6 should be denoted in more detail.

5. The differences and findings between experiments 1-8 should be described in more detail.

6. Ensure to review and correct any typos and grammatical errors throughout the manuscript.

Round 2

Reviewer 1 Report

Comments and Suggestions for Authors

The paper can be published in its current form.

Author Response

We thank the reviewer for this favorable assessment.